# Effect of an Oxygen-Based Mechanical Drug Delivery System on Percutaneous Permeation of Various Substances In Vitro

**DOI:** 10.3390/pharmaceutics14122722

**Published:** 2022-12-05

**Authors:** Anna-Lena Elksnat, Paula Zscherpe, Karina Klein, Jessika Maximiliane Cavalleri, Jessica Meißner

**Affiliations:** 1Department of Pharmacology, Toxicology and Pharmacy, University of Veterinary Medicine Hannover, Foundation, Bünteweg 17, 30559 Hannover, Germany; 2Musculoskeletal Research Unit, Vetsuisse Faculty, University of Zurich, Winterthurerstrasse 260, 8057 Zurich, Switzerland; 3Clinical Unit of Equine Internal Medicine, Department for Companion Animals and Horses, University of Veterinary Medicine Vienna (Vetmeduni), Veterinärplatz 1, 1210 Vienna, Austria

**Keywords:** permeation enhancer, transdermal drug delivery, Franz-type diffusion cell, porcine skin, skin absorption, oxygen

## Abstract

Transdermal drug administration is an elegant method to overcome various side effects of oral or parenteral drug administration. Nevertheless, due to an effective skin barrier, which is provided by the *stratum corneum*, transdermal drug delivery is sometimes very slow and ineffective. Thus, the effect of a medical device (DERMADROP TDA) for transdermal penetration of drugs in conjunction with a special vehicle emulsion on percutaneous permeation of several substances (with different physicochemical properties) was investigated in Franz-type diffusion cells with porcine skin over 28 h. This medical device disperses pharmaceutical agents via oxygen flow through an application system, which is used in conjunction with specially developed vehicle substances. Substance permeation of various substances with different physicochemical properties (diclofenac, enrofloxacin, flufenamic acid, indomethacin, and salicylic acid) was examined after application with a pipette and with the medical device. Therefore, acceptor media samples were collected up to 28 h after drug administration. Drug concentration in the acceptor medium was determined via high-performance liquid chromatography. Enhanced permeation was observed for diclofenac, enrofloxacin, flufenamic acid, indomethacin, and salicylic acid after oxygen-based administration. This correlates negatively with the molecular weight. Thus, drug administration can effectively be enhanced by a medical device using oxygen.

## 1. Introduction

Transdermal drug delivery represents an elegant method to overcome side effects after systemic administration and to maintain a constant drug flow into the skin. Nonetheless, the flux of a topically applied drug into the skin is limited due to a high permeability barrier, which is represented by the outmost layer of the skin, the *stratum corneum*. Thus, various approaches have been made to overcome the skin barrier in order to enhance transdermal drug delivery. On the one hand, chemical penetration enhancers which penetrate into skin to reversibly lower skin barrier resistance can be used (e.g., sulphoxides like dimethyl sulfoxide, alkanols like ethanol, or glycols like propylene glycol [1]). They may disrupt the intracellular keratin domains or desmosomal connections, alter metabolic activity in the skin, dissolve lipids from the *stratum corneum*, or influence the solubility of the drug in the vehicle [1,2,3]. On the other hand, a range of mechanical approaches have been developed in the last decades [4,5] to overcome the epidermal barriers. These techniques can be divided into two main areas: direct and indirect approaches. Direct approaches (e.g., microneedles, microdermabrasion) represent mechanical disruptors of the epidermis [4,6,7,8], while indirect approaches consist of laser or electrical instrumentation to create pores into the skin followed by drug application [4].

In recent years, a few studies have revealed a potential benefit of a novel mechanical permeation enhancer system based on oxygen-flow [9,10,11]. With this approach the non-steroidal drug carprofen and the folic acid antagonist methotrexate could successfully be delivered into the skin [11] and deeper layers [9,10], despite their high molecular weight or lipophilicity. Thus, our study was conducted to investigate the permeation efficacy of this oxygen-based drug delivery system in vitro with various test substances. An in vitro approach was used in order to avoid animal experiments in accordance with the 3R-principles [12]. Thus, Franz-type diffusions experiments were carried out in accordance with OECD guideline 428 [13]. Test substances were chosen with respect to differences in lipophilicity, molecular weight, and melting point, since physicochemical drug characteristics affect transdermal drug delivery in various species [14]. In conclusion, our study reveals the potential of this oxygen-based medical device for several drugs to effectively overcome the skin barrier.

## 2. Materials and Methods

### 2.1. Materials

All reagents used for the present study were of the highest purity available. Diclofenac, enrofloxacin, flufenamic acid, indomethacin, and salicylic acid were obtained from Sigma-Aldrich GmbH, Steinheim, Germany. Matrix-A TDA was kindly provided by Meddrop BioMedical Technologies GmbH, Hamburg, Germany. It consisted of aqua, isopropyl myristate, PEG-8, caprylic/capric glycerides, isopropyl alcohol, propylene glycol, urea, polyglyceryl-6 dioleate, phosphatidylcholine, alcohol denat., hydrolyzed sodium hyaluroante, sodium hyaluronate, and tocopherol. Methanol (MeOH) and acetonitrile (ACN) were obtained from Applichem GmbH, Darmstadt, Germany.

### 2.2. Buffers

The salts used for buffer production were obtained from Merck GmbH, Darmstadt, Germany. Phosphate buffered saline (PBS, pH 7.4) contained 0.2 g KCl, 8.0 g NaCl, 0.2 g KH_2_PO_4_, 1.44 g Na_2_HPO_4_ × 2H_2_O per liter deionized water. McIlvaine citrate buffer (MICP; pH 2.2) contained 20.8 g citric acid anhydrous, 0.4 g Na_2_HPO_4_ × 2H_2_O per liter deionized water. Citrate buffer (pH 3.0; 0.01 mol/L) contained 1.8 g citric acid monohydrate and 0.43 g trisodium citrate dehydrate per liter deionized water.

### 2.3. Transdermal Formulations

Due to different physicochemical properties of the drugs, the following solutions of the drugs were used dissolved (1:1) in Matrix A TDA: diclofenac (18.75 mg/mL deionized water), enrofloxacin (0.1 mg/mL PBS with 0.5% NaOH (1N)), flufenamic acid (0.7 mg/mL deionized water), indomethacin (1.4 mg/mL deionized water with 3% NaOH (1N), and salicylic acid (2.94 mg/mL deionized water with 5% NaOH (1N)).

### 2.4. Membranes

The skin was obtained from pigs directly after euthanasia (not related to this study; 5 months old; hybrid breeding; male and female, ethical approval code 33.8-42502-04-21/3760) at the Institute of Animal Nutrition, University of Veterinary Medicine Hannover, Foundation. Before excision, visual inspection was performed to ensure the integrity of the skin. Only intact and damage-free skin was excised. In brief, skin flaps were removed from lateral thorax and abdomen and were stored light and air protected at −20 °C until use (maximum 6 weeks). Prior to the experiment, the skin flaps were gently defrosted at room temperature and an electrical dermatome (Zimmer GmbH, Eschbach, Germany) was used to obtain 700 μm (±100 μm) thick split skin samples.

### 2.5. Medical Transdermal Application Device

DERMADROP TDA (Figure 1; Meddrop BioMedical Technologies GmbH, Hamburg, Germany) consisted of an oxygen supply and an application system. The high concentrated oxygen (in average 98%) served as a propellant, permitting droplets to reach the skin with a speed of approximately 265 m/s [11]. The oxygen flowed through a valve for pressure reduction and a treatment tube to the application device. The applicator represented a nanodispersion device. The pharmaceutical formulations were filled into the drug reservoir through a cartridge. The oxygen propelled and transported the carrier substance under pressure through the diffuser. Thus, due to the so-called Venturi effect (in response to a constricted area of flow the fluid pressure decreases), the microemulsion was atomized with droplets in the range of nanometers.

### 2.6. Franz-Type Diffusion Cell Experiments

Franz-type diffusion cells (effective diffusion area of 1.77 cm^2^) with a 12 mL acceptor chamber were used in this study. Porcine split skin was incubated for 30 min in phosphate buffered saline (PBS; pH 7.4) before mounting on the diffusion cells. The acceptor chamber was filled with PBS and a temperature of 34 °C was maintained during the whole experiment to ensure a skin sample temperature of 32 °C according to OECD guideline 428 [13]. Next, porcine split skin was placed on the acceptor chamber. A total volume of 16 µL of each test substance (as described in Section 2.3.) was applied (a) by using a pipette (control) or (b) via DERMADROP TDA (distance 1 cm to the skin surface). Acceptor samples (400 µL) were taken at the following times: 0, 1, 2, 4, 6, 22, 24, 26, and 28 h and were filled up with 400 µL PBS to maintain 12 mL acceptor volume. Each substance was examined in 6 animals with two technical replicates. Samples were stored at −20 °C up to 2 months before analysis.

### 2.7. High-Performance Liquid Chromatography (HPLC)

Diclofenac, flufenamic acid, indomethacin, and salicylic acid were analyzed in the acceptor media by validated UV-VIS high performance liquid chromatography methodologies using a Merck LiChroCART 125-4 Lichrospher 100 RP18e (5 mm) column (Merck GmbH, Darmstadt, Germany) and a LiChri-ART 4-4, LiChrospher 100 RP-18e, 5 µm precolumn (Merck GmbH, Darmstadt, Germany) [14]. A 168 UV-VIS detector from Beckman (Fullerton, CA, USA) was used at different conditions for substance analyses (Table 1; diclofenac, flufenamic acid, indomethacin, salicylic acid). For enrofloxacin analysis the following system was used [15]: a 508-autosampler (Beckmann, Germany), a 126-solvent system pump (Beckmann, Germany), a CC 250/4 NUCLEODUR 100-5 C18e, 25 cm column (Macherey und Nagel GmbH und Co. KG, Düren, Germany), and a LiChri-ART 4-4, LiChrospher 100 RP-18e, 5 µm precolumn (Merck GmbH, Darmstadt, Germany). Detection was performed with a fluorescence detector (RF-551, Shimadzu Torrance, Torrance, CA, USA; Table 1).

The conditions of analyses of each test substance are described in detail in Table 1. The content of each substance was determined using the external standard method [16].

### 2.8. Histological Examination

Histological sections were prepared using a cryostat at −20 °C. Slices of 10 µm thickness (transversal to skin surface) were prepared and were stained with hematoxylin eosin (HE) as described by Stahl et al. [17]. A light microscope (Leica, Camera AG, Wetzlar, Germany) was used in 40-fold magnification to analyze 10 slices. Therefore, each 10 areas were determined per slice. Thickness of the *stratum corneum* was determined using ImageJ 1.53 (NIH, Bethesda, MD, USA). Further histolopathological examinations of the viable skin layers, e.g., cell amounts in the dermis, cannot be determined as result of the different treatments, since skin samples were stored at −20° C up to 6 weeks before usage and represent non-viable membranes.

### 2.9. Data Analysis

The plot of the cumulated amount of substance versus time was used for calculating the apparent permeability coefficient (Papp) according to Niedorf et al. 2008 [15] as well as for determination of the lagtime and recovery. Values from replicated experiments (2 technical replicates per drug and treatment) in skin from the same individual were used as average. Differences in permeation parameters (P_app_ value, lagtime, and recovery) were analyzed using the Mann–Whitney Test (GraphPad Prism 9.01 (GraphPad Software Inc., San Diego, CA, USA)) with a significance level of *p* < 0.05.

## 3. Results

### 3.1. Skin Permeation

#### 3.1.1. Permeation Rates

Skin permeation profiles of all tested substances differed markedly between control (pipette) and DERMADROP TDA (Figure 2). Higher permeation rates were observed for all substances after application using DERMADROP TDA. The effect was only slightly visible within the first 6 h except for flufenamic acid and salicylic acid.

#### 3.1.2. Permeation Parameters

Higher P_app_ values were observed for substances applied via DERMADROP TDA in comparison to control. Significant differences in P_app_ values were found for all substances except flufenamic acid (Table 1). Concerning the lagtimes, lower lagtimes were detected for DERMADROP TDA applied substances except for flufenamic acid (12.23 h DERMADROP TDA vs. 9.28 h control) and indomethacin (2.09 h DERMADROP TDA vs. 1.14 h control), although no statistical significance was found between lagtimes after DERMADROP TDA application or pipette application in any substance (Table 2).

After 28 h, statistically significant differences in recoveries were observed for diclofenac, indomethacin, and salicylic acid (Table 3) between DERMADROP TDA treated samples and control samples. The quotient of recovery DERMADROP TDA to recovery control was the highest for salicylic acid (7), followed by flufenamic acid (4) to diclofenac, enrofloxacin, and indomethacin (2).

#### 3.1.3. Correlation between Permeation Parameters and Physicochemical Properties

The correlation between the enhancement of the P_app_ values and physicochemical properties (melting point, lipophilicity, and molecular weight) revealed a negative correlation to the molecular weight (R^2^ = 0.95). Thus, molecules with small molecular weights (e.g., salicylic acid, 138 g/mol) especially benefit from this medical device, since an 8-fold enhancement could be detected. Melting points and lipophilicities did not correlate with enhancement in P_app_ values (R^2^ = 0.21 and R^2^ = 0.01, respectively).

### 3.2. Histology

The histological examination of the *stratum corneum* revealed no differences between control and DERMADROP TDA treated skin samples (Figure 3), with a comparable thickness of the horny layer (13.96 vs. 13.17 µm; Table 4).

## 4. Discussion

Penetration enhancement represents a common method to overcome the skin barrier for effective transdermal drug delivery [1]. Various approaches are therefore used, depending on the applied drug [18]. While some methods damage the skin barrier to promote substance penetration (e.g., microneedling, tape stripping, abrasion) [6,7,19], other methods work with changes in the connectivity of the skin layers (e.g., chemical penetration enhancers like dimethyl sulfoxide or Azone) [1].

The method of oxygen-based substance application to the skin that we examined in the present study did not show any damage to the skin surface in histological images. Interestingly, all tested substances especially showed a penetration-promoting effect on the second day after substance application. A strong barrier disruption should rather be immediately effective, comparable to microneedle pretreatment [20]. This result is also supported by the calculated lagtimes, since the exposure time in the skin could not be shortened by oxygen-based application. Further investigations on repeated drug application on the skin could therefore be of interest for risk assessment as well as dosage regimes. Sidler et al. were able to show for the transdermally applied substance carprofen that joint defects had better healing rates through oxygen-based application in a stifle joint defect model in sheep [9,10]. In contrast to intravenously treated sheep, a modest accumulation of carprofen in plasma and synovial fluid was observed in the transdermally treated animals over the 6-week treatment period. Furthermore, Lebas et al. demonstrated an effective treatment option for superficial basal cell carcinoma or extramammary Paget disease by oxygen flux assisted methotrexate treatment [11]. In our study all test substances could be delivered into the skin more effectively by a factor of at least two by means of oxygen-based application, with better effects for small molecules with small molecular weights. It may be that small molecular weight substances are more effectively swirled and nanodispersed by the oxygen flow than substances with higher molecular weight [21], due to the so-called Venturi effect [9,10,11]. Thus, a negative correlation of substance permeability with molecular weight was observed in our experiments.

As a result, the absolute applied amount of applied substance can be reduced accordingly, which benefits a sustainable use of transdermally applied medicinal substances, since large parts of transdermally applied substances remain on the skin [22] and can accordingly get into the environment of the treated animals/humans [23] by washing of the skin/fur or clothing previously exposed to drugs.

No toxicity assessment was performed in our study, since we used non-viable skin samples in Franz-type diffusion chambers to gain information about the influence of the medical device on skin permeability barrier. However, looking at published in vivo studies in human and sheep after treatment with DERMADROP TDA [9,10,11], they all show a high level of skin tolerance in vivo. For translation from our in vitro study to in vivo application in animals and humans, all parameters like application volume, application time, and distance to the skin should be defined since they influence the effectiveness of the application. In addition, it could be interesting to determine the depth of penetration of the system into the skin. Therefore, e.g., a radiolabeled or fluorescence labeled substance could be applied in vitro.

In the present study only one type of vehicle was used for all investigated substances. Interestingly, independent of the lipophilicity or melting point enhancement of drug delivery was observed. The use of different carrier matrices would thus be interesting for future investigations, which could offer further potential for increasing the penetration rates due to the interactions with the applied substances. For example, Lebas et al. used vehicles (LP3 micro-emulsions) based on an oil-in-water or water-in-oil micro-emulsion for a successful methotrexate delivery into the skin [11]. Consequently, special carrier matrices should be used for different topically applied drugs according to their physicochemical drug characteristics in the future.

## 5. Conclusions

In summary, based on a series of in vitro evaluations, we demonstrated the potential of permeation enhancement of an oxygen-based medical device especially for substances with small molecular weight. The histological images indicate no noticeable damage in the skin. The findings of our study suggest that an oxygen-based drug flux to the skin represents a promising approach to enhance transdermal drug delivery.

## Figures and Tables

**Figure 1 pharmaceutics-14-02722-f001:**
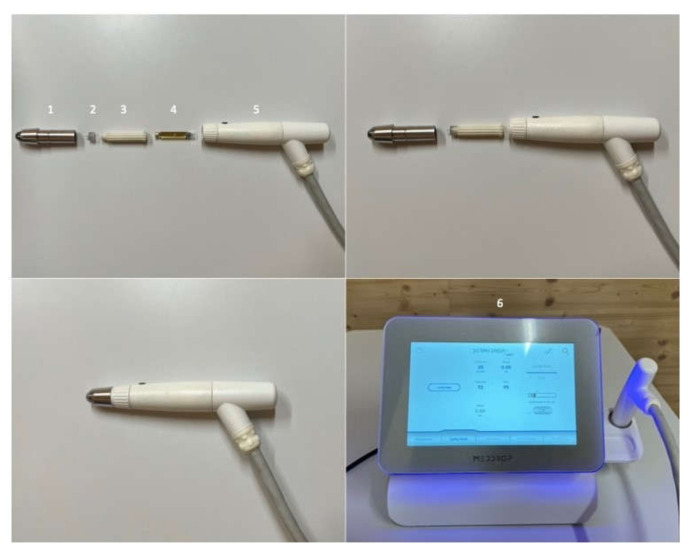
DERMADROP TDA system; (1) applicator tip, (2) TDA pen, (3) cartridge case, (4) cartridge, (5) applicator body, (6) DERMADROP device.

**Figure 2 pharmaceutics-14-02722-f002:**
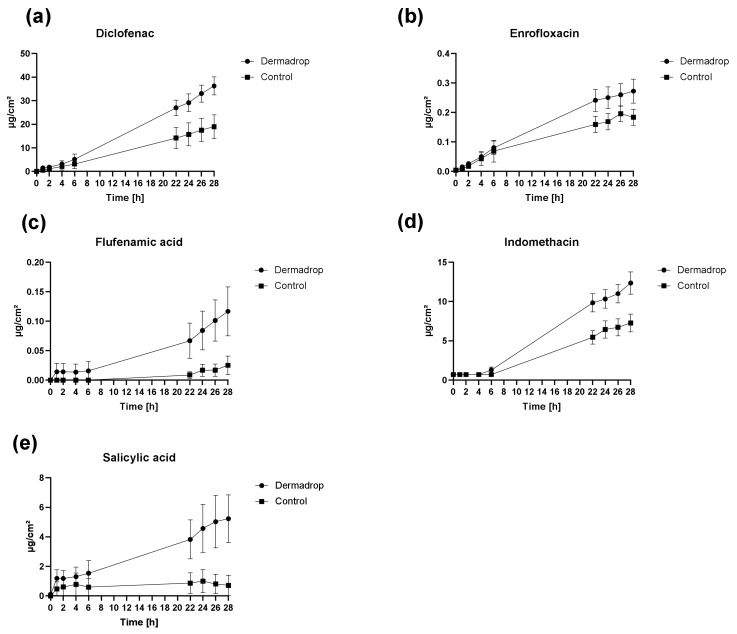
Cumulative amount of the different drugs after application via DERMADROP TDA and via pipette (control) over 28 h. (**a**) Diclofenac; (**b**) enrofloxacin; (**c**) flufenamic acid; (**d**) indomethacin; and (**e**) salicylic acid. Data are shown as mean ± SEM; *n* = 6.

**Figure 3 pharmaceutics-14-02722-f003:**
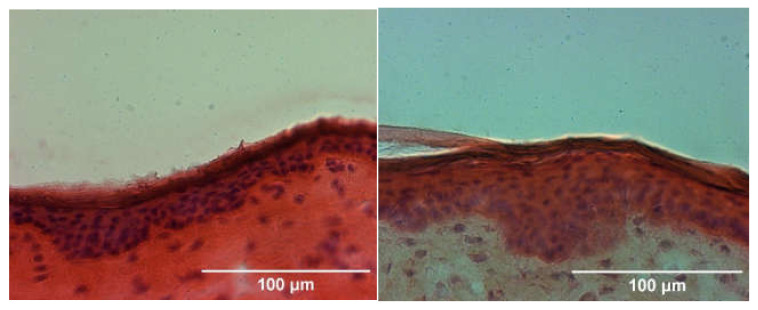
Histological section of porcine skin treated with 16 µL Matrix TDA via the medical device DERMADROP TDA (**left**) and pipette (**right**). HE-stained, 10 µm thin skin sections cut transversal to skin surface; bar represents 100 µm.

**Table 1 pharmaceutics-14-02722-t001:** Conditions of drug analysis (via HPLC) and physicochemical properties of the drugs; MICP = McIlvaine citrate buffer.

Substance	Wavelength (nm)	Injection Volume (µL)	Flow-Rate (mL/min)	Mobile Phase (*v*/*v*)	Limit of Detection (ng/mL)	Limit of Quantification (ng/mL)	Molecular Weight [g/mol]	Octanol-Water-Coefficient (logK_ow_)	Melting Point [°C]
Diclofenac	238	30	1.5	80:20 MeOH:MICP	5.5	20	296	4.51	289
Enrofloxacin	excitation 280/emission 450	30	1.5	85:15 Citrate buffer:ACN	0.2	0.6	359	0.70	220
Flufenamic acid	284	30	1.0	80:20 MeOH:MICP	14.4	50	281	4.67	134
Indomethacin	237	100	1.5	70:30 ACN:MICP	27.2	50	357	3.10	158
Salicylic acid	239	100	1.5	30:70 ACN:MICP	54.2	200	138	2.06	158

**Table 2 pharmaceutics-14-02722-t002:** P_app_ values and lagtimes of the different drugs applied via DERMADROP TDA and via pipette (control). Asterisks (*) indicate *p* < 0.05 (Mann–Whitney Test).

	P_app_ Values (cm/s)		Lagtime (h)	
	DERMADROP TDA	Control	DERMADROP TDA	Control
Diclofenac	4.42 × 10^−8^ ± 1.54 × 10^−8^ *	2.15 × 10^−8^ ± 1.24 × 10^−8^	2.58 ± 2.24	3.13 ± 1.89
Enrofloxacin	6.59 × 10^−8^ ± 2.03 × 10^−8^ *	3.98 × 10^−8^ ± 1.29 × 10^−8^	0.68 ± 1.16	1.32 ± 1.55
Flufenamic acid	7.45 × 10^−9^ ± 4.05 × 10^−9^	1.73 × 10^−9^ ± 2.49 × 10^−9^	12.23 ± 8.47	9.28 ± 10.90
Indomethacin	9.60 × 10^−8^ ± 3.40 × 10^−8^ *	5.42 × 10^−8^ ± 2.59 × 10^−8^	2.09 ± 2.17	1.14 ± 1.78
Salicylic acid	6.96 × 10^−8^ ± 1.16 × 10^−7^ *	9.25 × 10^−9^ ± 5.12 × 10^−9^	1.97 ± 0.97	2.42 ± 0.13

**Table 3 pharmaceutics-14-02722-t003:** Recoveries of the different drugs applied via DERMADROP TDA and via pipette after 28 h. Asterisks (*) indicate *p* < 0.05 (Mann–Whitney Test).

	Recovery (%)		Recovery DERMADROP TDA/
	DERMADROP TDA	Control	Recovery Control
Diclofenac	42.58 ± 14.40 *	22.26 ± 14.14	2
Enrofloxacin	62.08 ± 18.54	40.43 ± 14.66	2
Flufenamic acid	3.06 ± 3.01	0.79 ± 1.20	4
Indomethacin	89.13 ± 14.52 *	57.21 ± 21.68	2
Salicylic acid	39.15 ± 29.79 *	5.30 ± 12.64	7

**Table 4 pharmaceutics-14-02722-t004:** Thickness of the *stratum corneum* of control samples in comparison to the medical device DERMADROP TDA treated samples after HE-staining; no statistical significance was observed (Mann–Whitney Test); *n* = 10.

Thickness of the *Stratum Corneum* [µm]	
DERMADROP TDA	Control
13.96 ± 2.55	13.17 ± 2.02

## Data Availability

All data are provided in this manuscript.

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
