# Peer review of "Effect of an Oxygen-Based Mechanical Drug Delivery System on Percutaneous Permeation of Various Substances In Vitro"

_pharmaceutics, 2022, doi:10.3390/pharmaceutics14122722_

Round 1
Reviewer 1 Report
Kindly find the comments in the uploaded word document. Thank you.

Author Response
- Thank you very much for your valuable comments on our manuscript. We are very happy about your improvments and addressed each point of you (please see below).
The research article seems to be interesting. It talks about enhancing transdermal permeation of some drugs via the aid of oxygen flow, through a device named “DERMADROP TDA”. However, here are some comments that would be beneficial to the submitted manuscript:
- In the introduction section the following changes were made in accordance with the reviewers comments:
- line 35: “Transdermal drug delivery represents an elegant method to overcome side effects after systemical administration and to maintain a constant drug flow into the skin”. The word “systemical” is changed to “systemic”.
- line 43: “can alter metabolic activity in the skin, solve lipids from the stratum corneum or influence the solubility of the drug in the vehicle”. The word “solve” is changed to “dissolve”.
- lines 51-53: “With this approach the non-steroidal drug carprofen and the folic acid antagonist methotrexate could successfully delivered into the skin and deeper layers, despite high molecular weight or lipophilicity”. This sentence is rephrased into: “could successfully be delivered”…..”despite their high molecular weight”.
- In the materials section,
- line 68: “phophatidylcholine”, is corrected to “phosphatidylcholine”.
- The ethical approval code for the animal study is now provided: No. 8-42502-04-21/3760
- In section “2.5. Medical transdermal application device”, a “labelled diagram” of the DERMADROP TDA (Meddrop BioMedical Technologies GmbH, Hamburg, Germany) is now provided (line 108). Consequently, the figure numbers changed in the whole manuscript.
- In section “2.6. Franz-type diffusion cell experiments”, the reference of OECD guideline 428 was added to explain 32°C skin temperature.
Reviewer 2 Report
This manuscript investigates the percutaneous permeation of various substances with different physical properties by using an oxygen-based medical device. Several indirect transdermal absorption enhancement methods using mechanical approaches have been reported. The study was conducted using a Franz cell diffusion device in vitro, utilizing a known basic technique and a known representative drug, and is not novel. The quantity and quality of findings are also not sufficient for publication in this journal. The impact of this medical device on the skin permeation mechanism of these drugs should be discussed in more depth, including a negative correlation with molecular weight.
Author Response
Thank you very much for your helpful comment to improve the quality of our manuscript. According to your comments, we discussed the impact of this medical device on skin permation in more depth (line 242-245; „In our study all test substances could be delivered into the skin more effectively by a factor of at least 2 by means of oxygen-based application, with better effects for small molecules with small molecular weights. Maybe small molecular weight substances are more effectively swirled and nanodispersed by the oxygen flow than substances with higher molecular weight [21], due to the so-called Venturi effect [9-11]. Thus, a negative correlation of substance permeability with molecular weight was observed in our experiments.“). We hope, that after improvement of our manuscript with all revisions to reviewers 1-3, you find it sufficient for publication, since it represents an elegant method for painless transdermal drug application.

Reviewer 3 Report
1- Authors didn't explain why isolated skin was used? Is not it working in animal model ?
2- In “Medical transdermal application device”, could you please provide diagram, scheme or photograph to illustrate the design structure
3- Authors didn't discuss how to control the parameters used in the experiment if the result transferred later into human ?
4- After application, cytotoxicity was not investigated
5- The depth point can be reached by this system was not pointed out
6- In Fig.2 Image (left), histopathological evaluation revealed to presence eosinophilic structure in dermis layers. Authors have to explain presence of this evaluation
Author Response
Thank you very much for your helpful comment to improve the quality of our manuscript. According to your comments, we revised the following sections in the manuscript:
- Authors didn't explain why isolated skin was used? Is not it working in animal model ? à An explanation is now given in lines 55-58 („An in-vitro approach was used in order to avoid animal experiments in accordance with the 3R-principles [12]. Thus, Franz-type diffusions experiments were carried out in accordance with OECD guideline 428 [13].“)
- In “Medical transdermal application device”, could you please provide diagram, scheme or photograph to illustrate the design structure à an illustration is now given in Figure 1 (line 108).
- Authors didn't discuss how to control the parameters used in the experiment if the result transferredlater into human ? àThis comment raises the point that translation from ex vivo Franz chamber experiments to in vivo application on patients can be challenging because of the greater difficulty to standardize and control all parameters in the clinical setting. We agree with the reviewer that this challenge is evident. But with detailed description about the application volume, application time and distance to the skin comparable results can be generated. A comment is now given in the manuscript in lines 254-256 (“For translation from our in vitro study to in vivo application in animals and humans all parameters like application volume, application time and distance to the skin should be defined, since they influence the effectiveness of the application.“).
Furthermore, in line 120 the distance in our setup is now given, as well (“distance 1 cm to the skin surface“).
- After application, cytotoxicity was not investigated. à Cytotoxicity studies were not performed, since exised skin samples in Franz-type diffusion cell experiments cannot be used for that. But the study described by Lebas et al. and Sidler et al. provide no indication for cell toxicity and excellent tolerance. à an explanation is now given in the manuscript (lines 251-255; „No toxicity assessment was performed in our study, since we used non-viable skin samples in Franz-type diffusion chambers to gain information about the influence of the medical device on skin permeability barrier. However, looking at published in vivo studies in human and sheep after treatment with DERMADROP TDA [9-11], they all show a high level of skin tolerance in vivo.”)
- The depth point can be reached by this system was not pointed out à This is a relevant comment, but we have no information about depth profile of the system. We added a comment on that in line 256-260 („In addition, it could be interesting to determine the depth of penetration of the system into the skin. Therefore, e.g. a radiolabeled or fluorescence labeled substance could be applied in vitro.”).
6- In Fig.2 Image (left), histopathological evaluation revealed to presence eosinophilic structure in dermis layers. Authors have to explain presence of this evaluation à We added an explanation about this in lines 150-153: „Further histolopathological examinations of the viable skin layers, e.g. cell amounts in the dermis, cannot be determined as result of the different treatments, since skin samples were stored at -20° C up to 6 weeks before usage and represent non-viable membranes.“ Thus, this finding represents an artefact, which is not caused by our treatment, since the reasons for this difference in cell amounts in the pictures have their origin in the time before slaughtering.

Round 2
Reviewer 2 Report
The authors carefully responded to the reviewers' comments.
The revised paper has improved and reached an acceptable level.
Reviewer 3 Report
Manuscript has been revised point by point according to reviewer comments and It is more acceptable NOW.